# TopoGuide: a Curriculum-base Finetuning Framework for Topologically-Consistent 3D Molecule Generation

## Abstract

Equivariant diffusion models can generate high-quality 3D molecular geometries but often struggle with chemical validity due to a lack of explicit guidance from the 2D molecular graph. While prior works have alleviated this by adding graph-based information to the model's input, this often increases architectural complexity and slows inference. We propose a new finetuning framework that instills 2D topological awareness into pre-trained 3D generative models without altering their core architecture. Our method enforces consistency between the representations of a target 2D graph and a generated 3D structure within a shared embedding space, guided by a consistency loss. By applying our framework to state-of-the-art models, we demonstrate a significant improvement in topological accuracy and chemical validity while preserving the original model's high-quality geometry and inference efficiency. Codes will be available after reviewing process.

## 1 Introduction

*De novo* molecular design is a critical task in drug discovery and materials science, holding the potential to drastically accelerate the search for novel functional molecules. In recent years, the field has shifted from generating 1D SMILES strings or 2D graphs towards the direct generation of 3D molecular structures (Hoogeboom et al., 2022). This advance is driven by equivariant deep learning models that operate on atomic coordinates, allowing them to learn the intricate probability distribution of stable conformations and generate geometrically sophisticated structures that respect physical symmetries.

However, existing 3D generative models are often trained on geometric data alone, neglecting the rich chemical knowledge encoded in the 2D molecular graph. This focus on coordinates without explicit topological guidance leads to a critical deficiency: the models can produce physically plausible atomic arrangements that are chemically invalid, resulting in poor performance on metrics like atomic valency and stability. To address this, some works have sought to augment the input of diffusion models with explicit graph features (Huang et al., 2024; Hua et al., 2023). In these approaches, the 2D graph is often processed by a separate graph neural network whose outputs are injected as conditioning at each step of the denoising process. While effective at improving chemical fidelity, this strategy tightly couples the 2D and 3D components, complicating the model architecture and increasing inference time. This leaves a need for a more general and efficient strategy to imbue 3D models with 2D chemical awareness without requiring architectural redesign.

To bridge this gap, we introduce `TopoGuide`, a novel finetuning framework that solves two key technical challenges. First, to **avoid architectural complexity and inference overhead**, we instill 2D topological awareness by enforcing consistency between the latent embeddings of the target 2D graph and the generated 3D structure. This decouples the 2D guidance from the inference process, preserving the original model's efficiency. Second, to **ensure training stability** and prevent the strong geometric prior from being corrupted

by the new objective, we apply this consistency loss using a curriculum-based schedule. This strategy gently steers the generator to produce topologically correct structures without catastrophic forgetting of its geometric knowledge. The result is a method that provides precise scaffold control while fully retaining the generative quality of the foundational model.

Our primary contributions are threefold:

- We introduce a novel and lightweight finetuning framework that stably instills topological awareness into pre-trained 3D generative models without altering their core inference architecture.

- We propose a scheduled optimization strategy that gradually introduces a 2D-3D consistency loss, solving the challenge of training instability while effectively guiding the generator to adhere to a target molecular scaffold.

- We demonstrate the effectiveness of our framework on state-of-the-art models, including **EDM**, **GeoLDM**, and **SymDiff**, showing significant improvements in topological accuracy without compromising geometric quality.

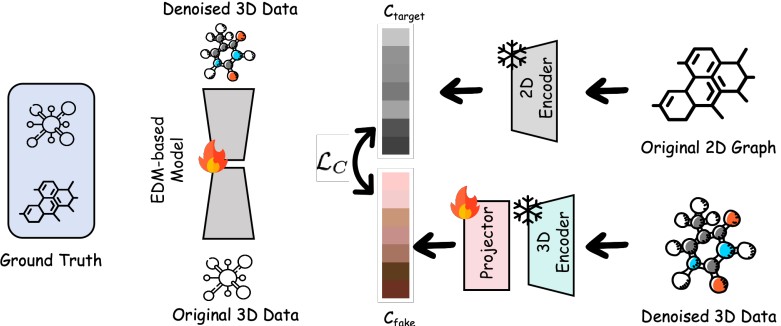

Figure 1: Overview of our proposed curriculum-based finetuning framework. The framework gradually introduces a 2D-3D consistency loss which includes a contrastive loss ($L_C$) and a reconstruction loss (not shown in the figure) via a scheduler to guide the unconditional 3D generator in learning topological structures while preserving geometric quality.

## 2  RELATED WORK

**Molecule Generation in 3D.** While extensive prior work focused on generating molecules as 2D graphs (Jin et al., 2018; Liu et al., 2018; Shi et al., 2020), recent interest has pivoted towards the direct generation of 3D molecular geometries. Early approaches in this domain were often autoregressive; models like G-Schnet and G-SphereNet (Gebauer et al., 2019; Luo & Ji, 2021) construct molecules by sequentially adding atoms or molecular fragments. This paradigm has been applied to structure-based drug design (Li et al., 2021; Peng et al., 2022; Powers et al., 2022), but it requires careful formulation of a complex action space and can be susceptible to compounding errors during the sequential generation process. Other distinct methods have utilized atomic density grids, which generate a molecule in a single step by predicting a density over a voxelized 3D space (Masuda et al., 2020). However, this approach lacks the desirable property of SE(3) equivariance and requires post-processing with a separate fitting algorithm.

**Diffusion models.** In a more recent development, denoising diffusion models have become the leading and most effective method for generating 3D molecules. These models function by learning to reverse a gradual corruption process applied to atomic coordinates and features. This reversal is commonly handled by 3D

equivariant neural networks, which maintain physical symmetries throughout the denoising process. This method has proven successful in several applications, such as targeted drug generation(Lin et al., 2022), antibody design(Luo et al., 2022), and protein design(Anand & Achim, 2022; Trippe et al., 2022). Despite their success, these models generally work directly with the original atomic coordinates.

## 3 PRELIMINARIES

### 3.1 2D AND 3D MOLECULAR ENCODERS

In molecular representation learning, molecules are commonly understood through two complementary views: a 2D topological graph and a 3D geometric structure. The 2D molecular graph represents atoms as nodes and bonds as edges, capturing the fundamental connectivity and topology of the molecule. Encoders for this view are typically 2D GNNs that operate via message passing to learn a representation of the molecule's topological features. In contrast, the 3D view incorporates the spatial coordinates of each atom, describing a specific low-energy spatial arrangement known as a conformer. This geometric information is vital as it relates directly to the molecule's potential energy surface, and a molecule's true properties are often a function of an ensemble of its possible conformers. 3D GNNs, which are often designed to be equivariant to rotations and translations, serve as encoders for this geometric view. Together, these two representations provide a comprehensive picture: the 2D graph defines the chemical blueprint, while the 3D geometry describes its physical realization in space.

### 3.2 THE PRINCIPLE OF SE(3) EQUIVARIANCE

A core requirement for generative models of 3D molecules is SE(3) equivariance, a property that embeds fundamental physical symmetries directly into the network architecture. This is necessary because the chemical and physical properties of a molecule are invariant to its position and orientation in space. These rigid-body transformations—rotations and translations—are mathematically described by the Special Euclidean group, SE(3).

A neural network is considered SE(3) equivariant if a transformation applied to its input results in an equivalent transformation of its output. For our denoising function, $\epsilon_\theta$, which takes atomic coordinates $\mathbf{x}$ and features $\mathbf{h}$ to predict a noise vector, this relationship is formally defined as:

$$\epsilon_\theta(\mathbf{R}\mathbf{x} + \mathbf{t}, \mathbf{h}) = \mathbf{R}\epsilon_\theta(\mathbf{x}, \mathbf{h}) \tag{1}$$

This equation states that if the input coordinates $\mathbf{x}$ are rotated by a matrix $\mathbf{R}$ and translated by a vector $\mathbf{t}$, the output noise vector is likewise rotated by $\mathbf{R}$. The atomic features $\mathbf{h}$, being intrinsic properties, are correctly treated as invariant. Imposing this constraint ensures that the model learns the fundamental geometry of molecules rather than memorizing pose-dependent features, which significantly improves data efficiency and generalization.

### 3.3 DIFFUSION MODELS

**Denoising Diffusion Models** are a type of generative model that create new data by learning to reverse a noise-adding process. The framework has two stages. First, a fixed forward process gradually adds random noise to a data sample over a series of steps until it becomes completely unrecognizable. Then, a neural network is trained for the reverse process, learning to systematically remove the noise at each step. By starting with a random noise sample and applying this learned denoising network iteratively, the model can generate a new, clean data sample.

**EDMs** are a specialized class of diffusion models designed for generating 3D data like molecular structures. Their key feature is SE(3) equivariance. This property means that if the input 3D structure is rotated or

translated in space, the model's output, such as the predicted noise vector, will be rotated or translated in the exact same way. This built-in understanding of physical symmetries allows the model to learn the fundamental geometric relationships of molecules more efficiently, as it doesn't need to see every possible orientation of a structure during training.

**Noise Prediction Loss**  The neural network in an EDM, denoted as $\phi$, is trained to predict the random noise $\epsilon$ that is added to a data point at a given timestep $t$. The noise vector $\epsilon$ consists of two components, $\epsilon = [\epsilon^{(x)}, \epsilon^{(h)}]$, corresponding to the atomic coordinates and features, respectively. A crucial step for maintaining translation invariance is to center the coordinate noise $\epsilon^{(x)}$ by subtracting its center of gravity. This prepared noise is then used to create a noised data sample $z_t = \alpha_t[x, h] + \sigma_t \epsilon$. The network's parameters are optimized by minimizing the mean-squared error (MSE) between the true noise and the network's prediction for that noise. The objective is to minimize the following loss function:

$$\mathcal{L} = \mathbb{E}_{x,\epsilon,t} \left[ \|\epsilon - \phi(z_t, t)\|^2 \right] \tag{2}$$

Here, the expectation is taken over the data, the centered noise, and the timestep. The loss measures the squared difference between the true noise $\epsilon$ and the network's prediction $\phi(z_t, t)$.

## 4  METHODS

Our framework finetunes a pre-trained 3D generative model to produce molecules that are topologically consistent with a target 2D graph. This is achieved by first aligning the latent spaces of pre-trained 2D and 3D encoders, and then using a topological consistency loss to guide the generative model.

### 4.1  MODEL COMPONENTS AND LATENT SPACE ALIGNMENT

Our framework utilizes several pre-trained components: a 2D graph encoder ($E_{2D}$), a 3D structure encoder ($E_{3D}$), and a 3D generative model backbone (e.g., EDM or GeoLDM) which is the target for finetuning. The 2D and 3D encoders are kept frozen.

A key challenge is that the pre-trained encoders may represent molecules in misaligned latent spaces. To address this, we introduce a lightweight **projector head**—composed of a multi-layer perceptron (MLP) and attention layers—attached to the output of the 3D encoder. Before finetuning the main generative model, we first train only this projector to map the 3D embeddings into the 2D encoder's embedding space.

### 4.2  TOPOLOGICAL CONSISTENCY OBJECTIVE

The core of our finetuning process is a topological consistency objective, $\mathcal{L}_{\text{consistency}}$, which is a composite loss inspired by the GraphMVP framework (Liu et al., 2022). It ensures that the latent representation of a generated 3D structure matches that of its corresponding 2D graph. This objective consists of two components: **A Contrastive Loss** ($\mathcal{L}_C$) encourages the model to pull the embeddings of corresponding 2D graph and 3D structure pairs (positive pairs) closer together in the latent space, while pushing apart the embeddings of non-matching pairs (negative pairs). **A Generative Loss** ($\mathcal{L}_G$) ensures that the latent embedding of one view (e.g., the 2D graph) contains sufficient information to reconstruct the latent embedding of the other view (e.g., the 3D structure). The final consistency loss is a weighted sum of these two components: $\mathcal{L}_{\text{consistency}} = \lambda_C \cdot \mathcal{L}_C + \lambda_G \cdot \mathcal{L}_G$.

### 4.3  MODEL-SPECIFIC FINETUNING PROTOCOLS

The topological consistency objective is integrated into the training of different generative models using tailored, multi-stage finetuning protocols.

For an EDM, the finetuning process directly updates the weights of its denoising network. The total loss function combines the model's original denoising loss with our topological consistency loss. To ensure stability, the consistency loss is introduced gradually using a curriculum learning schedule, which slowly increases its weight from zero. Our method successfully promote the model to learn the topological constraints without corrupting its strong, pre-trained geometric prior.

Both GeoLDM (Xu et al., 2023) and SymDiff (Zhang et al., 2024) require finetuning protocols tailored to their unique architectures, a little differing from the direct approach applied to EDM. For example, GeoLDM is a latent diffusion model with a VAE-based architecture. We adopt a two-stage finetuning process to effectively adapt it. First, we finetune only the VAE component of GeoLDM. The goal is to align its latent space with the chemical topology. This is done by training the VAE with the reconstruction loss and our topological consistency loss, while the diffusion model part remains frozen. After the VAE's latent space is aligned, we proceed to finetune the entire GeoLDM model end-to-end, including the latent diffusion network. This stage uses the combined VAE and diffusion losses, now guided by the topologically-aware latent space.

Table 1: Performance comparison with state-of-the-art 3D molecule generators on **QM9**. Our finetuned models (**++**) are evaluated against their original versions and other baselines. Higher is better (↑) for percentages. Lower is better (↓) for all distance metrics. **TV**: Total Variation Distance. $W_1$: Wasserstein-1 Distance. **Len**: Bond Length. **Ang**: Bond Angle.

| | Validity & Diversity | | | Distributional Similarity | | | | |
|---|---|---|---|---|---|---|---|---|
| Model | Valid %↑ | Stable %↑ | Unique %↑ | Atom TV↓ | Bond TV↓ | Valency $W_1$ ↓ | $W_1$ (Len)↓ | $W_1$ (Ang)↓ |
| *Reference Data* | *98.9* | *98.7* | *99.9* | *0.003* | *0.000* | *0.001* | *0.000* | *0.120* |
| VoxMol | 98.7 | 89.3 | 92.1 | 0.029 | 0.009 | 0.023 | 0.003 | 1.96 |
| FuncMol$_{dec}$ | 100.0 | 88.6 | 81.1 | 0.066 | 0.032 | 0.022 | 0.006 | 1.21 |
| FuncMol | 100.0 | 89.2 | 92.8 | 0.012 | 0.006 | 0.021 | 0.005 | 1.56 |
| EDM | 99.0 | 97.9 | 98.5 | 0.021 | 0.002 | 0.011 | 0.001 | 0.440 |
| **EDM++** | **99.2** | **98.0** | **98.7** | **0.019** | 0.002 | **0.008** | **0.001** | **0.421** |
| GeoLDM | **100.0** | **97.5** | 98.0 | 0.017 | 0.003 | 0.005 | 0.007 | 0.435 |
| **GeoLDM++** | 99.9 | 97.3 | **98.9** | **0.015** | 0.003 | **0.005** | **0.006** | **0.429** |
| SymDiff | 99.9 | 98.1 | 98.2 | 0.025 | 0.011 | 0.017 | 0.010 | 0.727 |
| **SymDiff++** | **100.0** | **98.5** | **98.4** | **0.011** | **0.008** | **0.006** | **0.008** | **0.595** |

## 4.4 FINETUNING PROCESS

The finetuning procedure updates the EDM's weights $\theta$ based on our composite loss, as detailed in Algorithm 1. For each training batch, we generate a 2D guidance vector $c_{target}$ from the input graph and compute two loss terms. The first is the standard denoising loss $L_{EDM}$ on a noised 3D sample. The second, a consistency loss $L_{2D\_consistency}$, requires a full generative pass to produce a 3D molecule and measures the distance between its resulting 3D embedding and the target 2D embedding. The weighted sum of these losses is then backpropagated to update the model's parameters.

---

**Algorithm 1:** Finetuning an Unconditional EDM with a Curriculum Schedule

---

**Input:** EDM $G_{\boldsymbol{\theta}}$, frozen encoders $E_{2D}, E_{3D}$, dataset $\mathcal{D}$, warmup steps $s_{\text{warmup}}$, final loss weights $\lambda_{C\_\text{final}}, \lambda_{G\_\text{final}}$.

**Output:** Finetuned EDM generator $G_{\boldsymbol{\theta}'}$.

$s \leftarrow 0$

Initialize optimizer $\text{Opt}_{\theta}$ for $G_{\boldsymbol{\theta}}$.

**for** *each epoch* **do**

    **for** *each batch of* $(G_{target}, \mathbf{x}_{real})$ *in* $\mathcal{D}$ **do**

        $\mathbf{c}_{\text{target}} \leftarrow E_{2D}(G_{\text{target}})$

        `// Part 1: Calculate Scheduled Loss Weights`

        $\text{ratio} \leftarrow \min(1.0, s/s_{\text{warmup}})$

        $\lambda_C(s) \leftarrow \lambda_{C\_\text{final}} \cdot \text{ratio}$

        $\lambda_G(s) \leftarrow \lambda_{G\_\text{final}} \cdot \text{ratio}$

        `// Part 2: Standard Denoising Step`

        $t \sim \text{Uniform}(1, T)$

        $\boldsymbol{\epsilon} \sim \mathcal{N}(0, I)$

        $\mathbf{x}_{\text{noised}} \leftarrow \sqrt{\bar{\alpha}_t}\mathbf{x}_{\text{real}} + \sqrt{1 - \bar{\alpha}_t}\boldsymbol{\epsilon}$

        $\boldsymbol{\epsilon}_{\text{pred}} \leftarrow G_{\boldsymbol{\theta}}(\mathbf{x}_{\text{noised}}, t)$

        $L_{\text{EDM}} \leftarrow \text{MSE}(\boldsymbol{\epsilon}, \boldsymbol{\epsilon}_{\text{pred}})$

        `// Part 3: Calculate Contrastive Loss on Predicted Clean Data`

        $\mathbf{x}_{0\_\text{pred}} \leftarrow (\mathbf{x}_{\text{noised}} - \sqrt{1 - \bar{\alpha}_t}\boldsymbol{\epsilon}_{\text{pred}})/\sqrt{\bar{\alpha}_t}$

        $\mathbf{c}_{\text{fake}} \leftarrow E_{3D}(\mathbf{x}_{0\_\text{pred}})$

        $L_C \leftarrow \text{Contrastive}(\mathbf{c}_{\text{target}}, \mathbf{c}_{\text{fake}})$

        $L_G \leftarrow \text{Generative}(\mathbf{c}_{\text{target}}, \mathbf{c}_{\text{fake}})$

        $L_{\text{consistency}} \leftarrow \lambda_C(s) \cdot L_C + \lambda_G(s) \cdot L_G$

        `// Step 4: Combine Losses`

        $L_{\text{total}} \leftarrow L_{\text{EDM}} + \lambda_T \cdot L_{\text{consistency}}$

        Update $\boldsymbol{\theta}$ using the gradient of $L_{\text{total}}$.

        $s \leftarrow s + 1$

    **end**

**end**

---

## 5 EXPERIMENTS

### 5.1 DATASETS

We evaluate our proposed framework on two standard benchmarks for 3D molecular modeling, **QM9** and **GEOM-Drugs**, to assess its performance on molecules of varying size and complexity.

- **QM9**: This dataset consists of approximately 134,000 small organic molecules with up to 9 heavy atoms (C, N, O, F). Each molecule is associated with a high-quality, ground-state 3D conformation calculated using density functional theory (DFT). Its comprehensive enumeration of small chemical space makes it ideal for validating the fundamental geometric and topological accuracy of our model.

- **GEOM-Drugs**: This is a significantly larger and more complex dataset containing over 300,000 drug-like molecules with an average of 44 atoms. The molecules in GEOM-Drugs are more repre-

sentative of real-world pharmaceutical compounds. Using this dataset allows us to test the scalability and applicability of our finetuning method on larger structures relevant to drug discovery.

### 5.1.1 Finetuning Protocol

We employ a **curriculum learning schedule** to stably introduce the consistency loss. The total objective is $L_{\text{total}} = L_{\text{EDM}} + \lambda_T \cdot (\lambda_C(s) \cdot L_C + \lambda_G(s) \cdot L_G)$. The weights $\lambda_C(s)$ and $\lambda_G(s)$ are ramped up from 0 to their final values, $\lambda_{C\_\text{final}}$ and $\lambda_{G\_\text{final}}$, over $s_{\text{warmup}}$ training steps using a linear schedule. We set the final weights $\lambda_{C\_\text{final}} = 0.1$ and $\lambda_{G\_\text{final}} = 0.4$, with a warmup period of $s_{\text{warmup}} = 50,000$ steps. All models are finetuned for a total of 200,000 steps using the Adam optimizer with a learning rate of $1 \times 10^{-5}$.

### 5.2 Evaluation Metrics

We assess the performance of our finetuned model using a comprehensive suite of metrics designed to measure the quality, realism, and diversity of the generated 3D molecules. These metrics are grouped into three main categories.

- **Validity and Stability**: This category evaluates whether the generated molecules are chemically correct. We report the percentage of **valid molecules** (`valid %`), which pass all RDKit sanity checks. We further measure the percentage of **stable molecules** (`stable mol %`) and **stable atoms** (`stable atom %`), which specifically check for correct atomic valencies. Higher percentages are better.

- **Distributional Similarity**: To measure realism, we compare the statistical distributions of generated molecules against the test set using the Wasserstein-1 ($W_1$) distance and Total Variation (TV) distance. A lower distance indicates that the generated molecules are more similar to the real ones. We measure:

    - **Chemical Accuracy**: The TV distance for **atom types** (`atom TV`) and **bond types** (`bond TV`), and the $W_1$ distance for the **valency** distribution (`valency $W_1$`).
    - **Geometric Accuracy**: The $W_1$ distance for the distributions of **bond lengths** (`bond len $W_1$`) and **bond angles** (`bond ang $W_1$`).

- **Diversity and Efficiency**: We report the percentage of **unique molecules** (`unique %`) among the valid generations to measure diversity.

In addition to these standard metrics, our primary evaluation for the finetuning process is **2D Topological Consistency**, which we define as the percentage of generated molecules whose inferred 2D graph is a perfect structural match (isomorphic) to the input 2D graph.

### 5.3 Baselines

To evaluate the effectiveness of our 2D-guided finetuning, we compare our method against several state-of-the-art 3D molecular generative models. For a fair comparison, we report the performance of both the original pre-trained models and our finetuned versions to directly measure the impact of our proposed method. Baselines include: **EDM**: The E(3) Equivariant Diffusion Model is our primary baseline. It is a diffusion-based model that generates molecules by directly denoising atomic coordinates and features in a manner that respects 3D symmetries. **GeoLDM**: A latent diffusion model that represents an alternative diffusion-based approach. It first encodes molecules into a compact latent space, performs the diffusion process there, and then decodes the results back into 3D structures. **VoxMol**: A generative model from a distinct architectural family. It operates by representing molecules on a 3D grid and using a voxel-based generation process. **SymDiff**: a novel method for constructing equivariant diffusion models via stochastic symmetrisation.

Table 2: Performance comparison of finetuned models (++ indicates our method) against their original versions on **Geom-Drugs**. Most of the metrics from QM9 are primarily reflective of small molecule properties which are not suitable for Geom-Drug dataset.

| Method | NLL $\downarrow$ | Atm. stability (%) $\uparrow$ | Val. (%) $\uparrow$ |
|---|---|---|---|
| *Reference Data* | — | *86.5* | *99.9* |
| GeoLDM | -198.28 | 84.4 | 99.3 |
| **GeoLDM+** | **-201.92** | **87.0** | **99.5** |
| EDM | -137.14 | 81.3 | 93.1 |
| **EDM+** | **-155.90** | **83.2** | **94.0** |
| SymDiff | -301.21 | 86.1 | **99.2** |
| **SymDiff+** | **-304.51** | **88.2** | 99.0 |

## 5.4 ABLATION STUDY

To quantify the individual contributions of the contrastive ($L_C$) and generative ($L_G$) components of our consistency loss, we conducted an ablation study on the EDM model by varying their respective weights, $\lambda_c$ and $\lambda_g$. Table 3 summarizes the performance on topological accuracy, as measured by Atom TV.

The study reveals a clear synergistic effect between the two loss components. Starting from a baseline Atom TV of 0.021 (when $\lambda_c = 0, \lambda_g = 0$), applying only the generative loss ($\lambda_c = 0$) improves the result to 0.020, while applying only the contrastive loss ($\lambda_g = 0$) provides a smaller improvement. The optimal performance is achieved when both losses are combined, reaching the lowest Atom TV of **0.019** at $\lambda_c = 0.1$ and $\lambda_g = 0.4$.

In addition, we studied the importance of our two main contributions: the latent space projector and the curriculum learning schedule. As shown in Table 4, removing the projector causes a significant drop in performance across all metrics, e.g. topological accuracy (Atom TV) worsening from 0.021 to 0.035. This highlights the necessity of the pre-alignment step for calibrating the 2D and 3D latent spaces.

Table 3: Ablation study of Atom TV values for different combinations of the loss weights $\lambda_c$ and $\lambda_g$. The best result (lowest Atom TV) is highlighted in bold.

| $\lambda_c$ | $\lambda_g$ | | | | |
|---|---|---|---|---|---|
| | **0** | **0.1** | **0.4** | **0.7** | **1** |
| **0** | 0.021 | 0.020 | 0.023 | 0.025 | 0.023 |
| **0.1** | 0.025 | 0.023 | **0.019** | 0.021 | 0.027 |
| **0.4** | 0.023 | 0.022 | 0.023 | 0.028 | 0.031 |
| **0.7** | 0.027 | 0.025 | 0.030 | 0.028 | 0.032 |
| **1** | 0.026 | 0.024 | 0.031 | 0.029 | 0.030 |

## DISCUSSION AND INSIGHTS

Our experiments show that `TopoGuide` significantly improves the topological accuracy and chemical validity of diverse 3D generative models while preserving their high-quality geometry. This confirms that a model's topological errors can be corrected efficiently through a post-hoc finetuning approach, avoiding the

need for complex architectural redesigns. Furthermore, our framework's effectiveness across different architectures highlights its model-agnostic nature. This versatility suggests a promising future direction: the same principle of latent space alignment could be used to guide generation based on other desirable chemical properties, positioning our work as a key step towards more controllable de novo molecule design.

**Insights: Global or Local Topology**  The enhancement from a 2D encoder may stem from its ability to provide a global topological constraint to a 3D model that excels at learning local geometry.

An EDM typically operates via a message-passing mechanism. This makes it exceptionally good at learning and enforcing local rules: an atom understands its immediate neighbors, leading to realistic bond lengths and angles. However, maintaining the correct overall structure of a large molecule with complex ring systems or long, flexible chains is a global problem. For a message-passing network, ensuring two distant parts of a molecule are correctly configured relative to each other requires propagating information across many steps, which can be challenging.

A 2D graph encoder, on the other hand, is specifically designed to create a single, holistic embedding that summarizes the entire molecule's global topology. This vector acts as a "fingerprint" for the entire scaffold. Therefore, our finetuning framework doesn't just re-introduce bond information the 3D model already has. Instead, it uses the 2D encoder's global embedding as an explicit scaffold-level constraint. The 3D model is still the expert at arranging atoms locally, but the 2D guidance acts as a global supervisor, ensuring the local decisions collectively assemble into the correct overall structure. This prevents errors where the local geometry is perfect, but the global topology is wrong.

Table 4: Ablation study of key framework components on the EDM model. We evaluate the performance degradation upon removing the projector and curriculum schedule.

| Model Variant | Validity & Diversity | | | Distributional Similarity | | | | |
|---|---|---|---|---|---|---|---|---|
| | Valid %↑ | Stable %↑ | Unique %↑ | Atom TV↓ | Bond TV↓ | Val. $W_1$↓ | $W_1$ (Len)↓ | $W_1$ (Ang)↓ |
| **EDM++** | **99.2** | **98.0** | **98.7** | **0.019** | 0.002 | **0.008** | **0.001** | **0.421** |
| EDM | 99.0 | 97.9 | 98.5 | 0.021 | 0.002 | 0.011 | 0.001 | 0.440 |
| w/o Projector | 91.3 | 87.0 | 94.1 | 0.035 | 0.007 | 0.020 | 0.009 | 0.491 |
| w/o Schedule | 90.2 | 82.0 | 91.0 | 0.041 | 0.010 | 0.025 | 0.011 | 0.510 |

## 6 CONCLUSION

In this work, we addressed the critical challenge in 3D de novo molecule generation—the frequent disconnect between geometric plausibility and topological validity—by introducing TopoGuide, a novel and efficient finetuning framework. Designed to instill 2D topological awareness into pre-trained equivariant diffusion models without altering their core architecture, our approach enforces consistency between the latent representations of a target 2D molecular graph and a generated 3D structure. By leveraging a multi-task objective composed of a contrastive loss and a generative loss introduced via a curriculum learning schedule, our framework effectively guides the generator to produce molecules that are both geometrically sound and topologically identical to a given scaffold. We demonstrated the efficacy and generality of TopoGuide by applying it to several state-of-the-art models, including EDM, GeoLDM, and SymDiff, showing a significant improvement in chemical validity and topological accuracy on the QM9 and GEOM-Drugs. Crucially, these gains were achieved while preserving the high-quality geometry generation and inference efficiency of the original models, offering a flexible and computationally inexpensive strategy for adding precise scaffold control to the powerful domain of 3D generative models.

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

## AI USAGE NOTIFICATION

The authors acknowledge the use of AI tools for minor assistance in polishing the writing of this paper. Specifically, an AI writing assistant was utilized to improve the clarity, grammar, and overall readability of the text. This tool was used solely to refine existing content and did not contribute to the conceptualization, research, or development of the scientific ideas presented in this work. All technical and scientific contributions are the original work of the authors.

## LOSS FORMULATION

The topological consistency objective used in our work is adapted from the GraphMVP framework (Liu et al., 2022). This framework leverages the natural correspondence between a molecule's 2D topological graph and its 3D geometric structure as two complementary "views" for self-supervised learning. The total loss is a synergistic combination of two distinct objectives: a contrastive loss and a generative loss.

**Contrastive Self-Supervised Loss ($\mathcal{L}_C$)** To learn discriminative features, we employ a contrastive loss that operates by comparing different molecules, thus leveraging an *inter-molecule* signal. For this task, a "positive pair" is defined as the 2D topological graph and a corresponding 3D conformer of the same chemical entity. Conversely, "negative pairs" are constructed by pairing the 2D view of one molecule with the 3D view of another. The training objective is to minimize the distance between the latent embeddings of positive pairs while simultaneously maximizing the distance for negative pairs. This forces the encoders to generate consistent representations for a given molecule, irrespective of the input view.

**Generative Self-Supervised Loss ($\mathcal{L}_G$)** To ensure the learned representations are information-rich, we use a generative loss in parallel with the contrastive one. This component operates on an *intra-molecule* basis, focusing on the relationship between the two views of a single molecule. The goal is to confirm that the latent representation of one view contains sufficient information to generate the representation of the other. As direct reconstruction from a continuous embedding to a discrete graph or 3D structure is challenging, we adopt the **Variational Representation Reconstruction (VRR)** strategy from GraphMVP. VRR sidesteps this issue by performing the reconstruction task entirely within the continuous latent space. This approach encourages the creation of comprehensive embeddings and effectively models the one-to-many mapping from a single molecular graph to its ensemble of possible 3D conformers.

Our total consistency objective is therefore a weighted combination of these two losses. This multi-task approach leverages the complementary strengths of the contrastive loss in discriminating between molecules and the generative loss in creating information-dense representations.

