# OpenReview forum: "TopoGuide: A Finetuning Framework for Topologically-Consistent 3D Molecule Generation"
_ICLR.cc/2026/Conference — ICLR 2026 Conference Withdrawn Submission_

### Official Review · Reviewer_c6RX · 2025-10-31

**Soundness:** 3
**Presentation:** 3
**Contribution:** 3
**Rating:** 8
**Confidence:** 3

**Summary:**

The paper introduces TopoGuide, a finetuning framework for that instills 2D topological awareness to 3D molecular generation algorithms by enforcing consistency between the latent embeddings of the target 2D graph and the generated 3D structure, decoupling guidance from inference, and applying a consistency loss using a curriculum-based schedule to ensure training stability. The purpose of this framework is to improve validity of 3D molecular generation algorithms by including important graph-based (2D) information when finetuning pretrained 3D models. The authors demonstrate the ability of this finetuning scheme to meaningfully improve validity and diversity of generated models while minimizing distributional drift from the target set of molecules.

**Strengths:**

The method introduced here is novel, useful and addresses an important gap in the field of 3D molecular generation. Importantly, it is a lightweight solution to addressing this problem. The authors demonstrate well the advantages conferred by the method to instill 2D topological awareness while maintaining the high-quality geometry generation and inference efficiency of the methods it is applied to.

**Weaknesses:**

While the method is promising, it has only been demonstrated on a few models, and performance on those models is not consistently better. It would be useful to do include a few additional models to see how performance improves, and also include metrics from a SOTA 2D method to help frame why the 3D + 2D-aware finetuning does confer improvements over 2D alone.

**Questions:**

I would like to see how the validity of molecules, and distributional shift, change over the course of fine tuning. Is it possible to show these metrics over finetuning steps?

Is it possible to include a 2D method as a baseline, or additional baseline models here that might be relevant, to really highlight how much this finetuning method improves performance over not just the base model itself but other topologically aware methods that might be SOTA.

---

### Official Review · Reviewer_2TsX · 2025-10-31

**Soundness:** 1
**Presentation:** 2
**Contribution:** 2
**Rating:** 2
**Confidence:** 4

**Summary:**

This paper proposes a new finetuning framework for pre-trained 3D molecule generative models (with a focus on diffusion models) with the goal of improving the 2D graph properties of the generated 3D molecules, e.g., chemical validity. This is achieved by finetuning the models using a consistency loss between the representations of a 2D graph and a generated 3D molecule in a shared embedding space.

**Strengths:**

* The problem of 3D molecule generative models struggling in 2D graph metrics is well-known and practically relevant.
* The proposed method of finetuning pre-trained models using target 2D graphs is promising, especially because it does not change their architecture nor their inference runtime.
* The paper is clear and well-written (but many details are missing, see weaknesses).

**Weaknesses:**

* The main contribution of this paper, which is the finetuning consistency loss, is not sufficiently explained and seems to be very similar to the pre-training loss used in GraphMVP [1], limiting the originality of the paper. Specifically, the exact loss equation is not provided, and it is unclear which parts of the GraphMVP framework are being adopted here.
* Many other details are missing. For example, how the projector head is trained (Section 4.1), how the tailored finetuning protocols for GeoLDM and SymDiff are implemented (Section 4.3). Code is not provided to verify the actual implementation.
* The authors claim their method achieves significant improvement in topological accuracy, while the reported results (Tables 1 & 2) show only minor improvement, which might not even be statistically significant, as no standard deviations for the metrics are reported.

**References**

[1] Liu, Shengchao, et al. "Pre-training molecular graph representation with 3d geometry." arXiv preprint arXiv:2110.07728 (2021).

**Questions:**

1. Can the proposed framework be used to train 3D molecule diffusion models from scratch (instead of finetuning pre-trained ones)? It would be a good addition to the paper to evaluate this setting.
2. In Equation 1, shouldn't the output also be translated by $t$ if we are talking about translation invariance?
3. In Table1, what does Stable% refer to? molecule or atom stability?
4. What is the role of $\lambda_T$ and how is it chosen?
5. Where do you report the 2D topological consistency metrics discussed in lines 314-316?
6. In Line 351, it states, "applying only the contrastive loss (λg = 0) provides a smaller improvement." However, the first column of Table 3 shows that there is no improvement. Can you explain?
7. Lines 383-388 suggest that EDM uses a message-passing GNN that operates on some graph structure; however, EDM uses the fully-connected graph of 3D molecules and therefore does not have the mentioned problems. Can the authors elaborate on this point?
8. Can you provide more details about the efficiency of the proposed approach, especially compared to the full training of the model?

---

### Official Review · Reviewer_ub4Q · 2025-11-01

**Soundness:** 2
**Presentation:** 1
**Contribution:** 1
**Rating:** 2
**Confidence:** 5

**Summary:**

This paper proposes TopoGuide, a finetuning framework designed to improve the chemical validity and topological accuracy of pre-trained 3D molecular generative models, particularly equivariant diffusion models (EDMs). The core idea is to enforce consistency between the latent embeddings of a target 2D molecular graph and a generated 3D structure without altering the base model's architecture. This is achieved through a composite loss function, adapted from self-supervised learning, which includes contrastive and generative components. The consistency loss is introduced gradually via a curriculum learning schedule to ensure training stability. The authors apply TopoGuide to several models (EDM, GeoLDM, SymDiff) and report improved performance on topological metrics while preserving geometric quality on the QM9 and GEOM-Drugs datasets.

**Strengths:**

1. The motivation is strong and highly relevant. Developing methods to instill chemical awareness into powerful 3D generative models is a critical research direction.
2.  The idea of a lightweight, post-hoc finetuning framework is appealing. In principle, such a method could offer a computationally efficient way to "correct" powerful pre-trained models without the need for a full retraining or a complex architectural redesign. The model-agnostic nature of the proposed framework is also a laudable goal.

**Weaknesses:**

1. The central overview figure (Figure 1) fails to clearly illustrate the proposed method. The relationships between key components are ambiguous.
2. The paper dedicates an excessive amount of space (Section 3) to preliminary discussions of well-known concepts like SE(3) equivariance and diffusion models. In contrast, the core methodology (Section 4) is rushed and lacks crucial technical details. Key components, such as the architecture of the "projector head" and the precise mathematical formulation of the contrastive and generative losses, are only described at a high level. This imbalance leaves the novel aspects of the work feeling underdeveloped.
3. The evaluation relies entirely on aggregate statistical metrics. For a generative model claiming to improve molecular topology and geometry, this is insufficient. The paper provides no visualizations of the molecules it generates. To substantiate its claims, it is essential to show examples of topologically incorrect molecules produced by a base model alongside the corrected, valid structures generated by the finetuned model. Without this qualitative evidence, the reader cannot verify the practical impact of the method or build intuition about the types of errors it addresses.
4. The experimental setup is flawed. TopoGuide is only benchmarked against unconditional 3D generators. The paper omits comparison to models such as joint 2D-3D generators or other graph-to-conformer methods.

**Questions:**

1. Could you please provide a revised Figure 1 that clearly and unambiguously illustrates the data flow, model interactions, and gradient updates during the finetuning process? Additionally, could you expand Section 4 with precise architectural details for the projector head and the full mathematical formulations for the consistency losses, while significantly condensing the preliminary background in Section 3?
2. To substantiate your claims of improved topological accuracy, would it be possible to add visualizations to the paper?
3. Could you please provide a justification for omitting comparisons to some joint 2D-3D generators? To properly contextualize your results, a comparison against these more relevant methods seems essential.

---

### Official Review · Reviewer_bHV7 · 2025-11-03

**Soundness:** 2
**Presentation:** 2
**Contribution:** 2
**Rating:** 2
**Confidence:** 4

**Summary:**

This paper proposes TopoGuide, a post-hoc finetuning framework designed to enforce 2D-3D topological consistency in pretrained 3D molecule diffusion models (e.g., EDM, GeoLDM, SymDiff). Instead of modifying the original generator or adding extra inference-time inputs, TopoGuide freezes the pretrained encoders and introduces a lightweight projector to align 3D latent embeddings with 2D molecular graph embeddings.
The method combines a contrastive loss and a reconstruction loss to bring the 3D representation closer to its corresponding 2D topology, with a curriculum-based warm-up to gradually increase the topological consistency loss weight.
Experiments on QM9 and GEOM-Drugs datasets show that TopoGuide improves chemical validity, 2D-3D isomorphism, and geometric accuracy across multiple pretrained models, without introducing additional inference cost.

**Strengths:**

The paper tackles a well-known issue in 3D molecule generation—geometrically plausible but topologically incorrect outputs—by providing a plug-and-play solution that does not alter model architecture or inference.

The approach can be directly adopted in molecular design workflows to improve scaffold fidelity without retraining large generative models.

**Weaknesses:**

The main innovation lies in combining latent-space alignment and curriculum finetuning. While practical, this could be viewed as an engineering refinement rather than a fundamentally new generative modeling framework.

The paper compares finetuned models (e.g., EDM++) against their original versions but omits comparison with models that explicitly condition on 2D molecular graphs during inference. This limits the fairness of the evaluation.

Figure 1 is hard to understand; from left to right, there are three isolated blocks. What is the connection between them?

The paper only provides two known formulas (Eq(1) and Eq(2)) in the PRELIMINARIES section, and offers no expressions for its own methodology, especially for the loss function. A mere textual description is insufficient. How does L_G “ensure that the latent embedding of one view (e.g., the 2D graph) contains sufficient information to reconstruct the latent embedding of the other view (e.g., the 3D structure)”.

**Questions:**

See Weaknesses.

---

### Note · Authors · 2025-11-19

**Comment:**

We sincerely appreciate all the reviewers for the time and dedication spent reviewing our submission. Every comment provided was deeply considered and proved highly valuable for the refinement of this work.

In the early stages of this project, we explored several training strategies that unfortunately failed to yield results. Once our proposed molecular generation framework finally achieved promising outcomes, I was eager to submit, which regrettably led to insufficient care in the final writing and presentation (e.g. valid concerns mentioned from Reviewers ub4Q and bHV7 regarding the visual clarity of Figure 1, the point raised by Reviewer 2TsX about the missing implementation details of our baselines, etc.). We acknowledge and appreciate the detailed concerns carefully raised from every reviewer. Given the high pressures and occasional difficulties of the current iclr2026 review cycle, we are truly grateful for the depth and seriousness with which each reviewer approached our paper. Your detailed feedback is deeply inspiring and provides a clear roadmap for our next iteration.

We extend a special thank you to Reviewer c6RX. Your acceptance score and recognition of our work’s potential provide immense motivation. For us navigating this new domain of AI4Science, this encouragement is invaluable.

After careful reflection on the comprehensive feedback and the current state of our manuscript, we have decided to withdraw this submission. We believe this will allow us the necessary time to address all structural and clarity issues thoroughly, and we do not wish to occupy any more of the reviewers' valuable time during the discussion phase. Thank you once again for your professional guidance.

Sincerely,

The Authors

**Withdrawal Confirmation:**

I have read and agree with the venue's withdrawal policy on behalf of myself and my co-authors.